# Commonly Prescribed Anticoagulants Exert Anticancer Effects in Oral Squamous Cell Carcinoma Cells In Vitro

**DOI:** 10.3390/biology11040596

**Published:** 2022-04-14

**Authors:** Li-Qiao R. Ling, Zichen Lin, Rita Paolini, Camile S. Farah, Michael McCullough, Mathew A. W. T. Lim, Antonio Celentano

**Affiliations:** 1Melbourne Dental School, The University of Melbourne, 720 Swanston Street, Carlton, VIC 3053, Australia; l.ling@student.unimelb.edu.au (L.-Q.R.L.); zichenl1@student.unimelb.edu.au (Z.L.); rita.paolini@unimelb.edu.au (R.P.); m.mccullough@unimelb.edu.au (M.M.); mathew.lim@alfred.org.au (M.A.W.T.L.); 2Australian Centre for Oral Oncology Research & Education, Perth, WA 6009, Australia; camile@oralmedpath.com.au; 3Oral, Maxillofacial and Dental Surgery, Fiona Stanley Hospital, Murdoch, WA 6150, Australia; 4Anatomical Pathology, Australian Clinical Labs, Subiaco, WA 6008, Australia; 5Dental Services, Alfred Health, Melbourne, VIC 3004, Australia

**Keywords:** oral cancer, oral squamous cell carcinoma, anticoagulants, heparin, warfarin, edoxaban, apixaban, rivaroxaban, dabigatran

## Abstract

**Simple Summary:**

Oral squamous cell carcinoma (OSCC) is the sixth most common cancer worldwide with 840,000 new cases and 420,000 deaths in 2020. Anticoagulants are widely prescribed medications routinely administered to help prevent blood clots. Despite the great relevance of these two topics, there is complete lack of knowledge regarding the potential effects that these drugs could exert on oral cancer patients. In this in vitro study, we comprehensively investigated the effect of anticoagulants on OSCC activity. This includes the effect of these drugs on cancer cell ability to survive, migrate to colonise distant sites, and resist treatment with conventional chemotherapy. We have demonstrated for the first time that various anticoagulants have anticancer effects on OSCC. Moreover, some of the anticoagulants tested were able to reduce the migratory ability of cancer cells. Finally, the great majority of anticoagulants studied reduced the effectiveness of the tested chemotherapeutic agent, allowing an increase in cancer cell proliferation. Our results highlight the need for urgent further research in the field, to improve the anticoagulant strategies in patients with oral cancer, and in turn their prognosis.

**Abstract:**

Oral squamous cell carcinoma (OSCC) is the most common head and neck cancer. With anticoagulant usage on the rise, it is important to elucidate their potential effects on tumour biology and interactions with chemotherapeutics. The aim of the present study was to investigate the effects of anticoagulants on OSCC cell lines and their interactions with the drug 5-fluorouracil (5-FU). Cell proliferation was assessed using an MTS in vitro assay in two human OSCC cell lines (H357/H400) and in normal oral keratinocytes (OKF6) treated with the 5-FU (0.2/1/5/10 μg/mL), conventional anticoagulants warfarin (1/5/10/20 μM) and heparin (5/20/80 U), as well as four new oral anticoagulants, dabigatran (5/10/20 μM), rivaroxaban (5/10/20 μM), apixaban (0.1/1/5 μg/mL), and edoxaban (5/10/20 μM). Cell migration was assessed at 3 h intervals up to18 h using a wound healing assay. Our results clearly demonstrate, for the first time, that commonly prescribed anticoagulants exert in vitro antiproliferative effects on OSCC cells. Furthermore, treatment with some anticoagulants reduced the migration of OSCC cell lines. Nevertheless, most of the anticoagulants tested reduced the effectiveness of the chemotherapeutic agent tested, 5-FU, highlighting potential flaws in the current pharmacological management of these patients. Our findings showed the need for the immediate translation of this research to preclinical animal models.

## 1. Introduction

Oral squamous cell carcinoma (OSCC) is the most common head and neck cancer, accounting for 90–96% incidence of all head and neck cancers [1]. Current strategies for OSCC treatment include surgery, radiation therapy, and adjuvant therapy, such as chemotherapy. 5-fluorouracil (5-FU) is a standard chemotherapeutic adjunct in OSCC treatment [2]. Globally, the overall 5-year survival of OSCC has not improved significantly beyond 50% and less than 1 in 5 patients who present with metastatic disease at the time of diagnosis will survive 5 years. The survival rate increases instead to 67% for early stage localised cancers [3]. Nevertheless, survival outcomes only capture part of the impact of OSCC, as individuals who survive their cancer experience overwhelming functional deterioration with attendant poor quality of life outcomes due to treatment sequelae, especially for later-stage disease. In fact, mouth cancer treatments are associated with profound debilitation and disfigurement, with patients living with chronic functional impairment (speech and swallowing).

Interest has grown in the potential of anticoagulants for treating/preventing cancer, due to links between coagulation and cancer biology and prognosis. Certain angiogenic processes are favoured in carcinogenesis, due to cell-to-cell interactions, localised hypoxia, and the expression of particular cytokines and growth factors [4,5]. 

Anticoagulants are widely prescribed medications, routinely administered to help prevent blood clots. They are administered to people at high risk of blood clots, to reduce their chances of developing serious conditions such as stroke and heart attack. This may include people with atrial fibrillation or an irregular heartbeat. Anticoagulants may also be prescribed to people who have had major surgery, such as aortic valve replacement, or those with certain blood disorders. The most prescribed anticoagulant is warfarin, and for short-term reversible use, heparin. Overall, Vitamin K antagonists have been successfully used over the last five decades; however, more recently, newer types of anticoagulants have become available and are becoming increasingly common. These new oral anticoagulants (NOACs) include rivaroxaban, dabigatran, apixaban, and edoxaban. Warfarin and NOACs are taken as tablets or capsules, while heparin can be given by injection. The new types of anticoagulants have some advantages compared to vitamin K antagonist, e.g., they target a specific single factor of the coagulation cascade rather than inhibiting the whole vitamin K synthesis. Nevertheless, even if NOACs are becoming more popular, they have not fully replaced the use of warfarin and heparin presently.

Recently, a study by Khan et al. systematically reviewed the use of new oral anticoagulants and direct oral anticoagulants in malignant patients [6]. A total of 12,269 patients were observed (64.19% presenting with active cancers vs. 35.80% observed as a control group). About 61.14% were using NOACs, 42.83% were on warfarin, and 2.72% were on low molecular weight heparin (LMWH). The NOACs used by these patients included edoxaban (6.81%), apixaban (5.28%), dabigatran (10.09%), and rivaroxaban (10.02%). In this study, the authors were able to stratify the patients following the anticoagulant regimen, concluding that drug interactions between anticancer drugs and NOACs are still not fully understood.

With anticoagulant usage on the rise, it is important to elucidate their potential effects on tumour biology and interactions with chemotherapeutics. However, little is known about these drugs’ effect on OSCC behaviour. Recently, we systematically reviewed the literature about this crucial topic, and have shown that there is a significant dearth of information surrounding interactions between anticoagulant therapy and OSCC [7]. The study by Ueda et al. [8] was the only one to investigate the effects of unfractionated heparin (UFH) on OSCC cell growth and apoptosis, and the mechanisms underlying its actions.

The aim of the present study was to investigate the in vitro effects of anticoagulants on OSCC cell lines and their interactions with the drug 5-FU. Our results have clearly demonstrated, for the first time, that commonly prescribed anticoagulants exert in vitro antiproliferative effects on OSCC cells. Furthermore, treatment with some anticoagulants reduced the migration of OSCC cell lines. Nevertheless, most of the anticoagulants tested reduced the effectiveness of the chemotherapeutic agent tested, 5-FU, unveiling a whole new dimension of chemotherapy resistance with dramatic potential clinical implications. Our findings showed the need for the immediate translation of these experimental conditions from bench to preclinical animal models.

## 2. Materials and Methods

### 2.1. Cell Lines

Two human OSCC cell lines derived from different intra-oral sites, H357/H400, and one normal human oral keratinocyte cell line, OKF6, were selected for this study. The OSCC adherent cell lines were established at Bristol Dental School, University of Bristol, UK, from primary explants of tongue (H357) and alveolar process (H400) squamous cell carcinoma [9]. All OSCCs were HPV-negative and were authenticated prior to commencing the experiments. All the cell lines/strains were derived prior to 2001 and, therefore, were not subject to Ethical Committee approval in the UK.

### 2.2. Culture Conditions

The OSCC cell lines were cultured as previously described [10]. Cell lines were cultured in 100 mm Petri plastic dishes (Corning 430167, Sigma-Aldrich, Castle Hill, NSW, Australia) and grown to 60–80% confluence before being further subcultured. OSCC cells were cultured using Dulbecco’s modified Eagle’s medium (DMEM) (D5796) and nutrient mixture F-12 Ham (N6658) in a 1:1 ratio (Sigma-Aldrich, Australia), supplemented with 10% foetal bovine serum (FBS) (SFBS-F, Bovogen, Keilor East, VIC, Australia), 1% penicillin streptomycin mixture (P4333, Sigma-Aldrich, Castle Hill, NSW, Australia), and 0.5 μg/mL hydrocortisone (HC) (H6909, Sigma-Aldrich, Castle Hill, NSW, Australia) in a humidified atmosphere at standard conditions (5% CO_2_, 37 °C). The normal human oral mucosal epithelial cell line, OKF6, was instead cultured in 100 mm Petri plastic dishes (Corning 430167, Corning, NY, USA) and grown to 60–80% confluence before being further subcultured. OKF6 cells were cultured using keratinocyte serum-free medium (K-SFM) (#17005-042, Thermo Fisher Scientific, Scoresby, VIC, Australia) containing 25 μg/mL bovine pituitary extract and 0.2 ng/mL human recombinant epidermal growth factor (as per manufacturer’s instructions), 0.4 mM CaCl_2_, 1% penicillin streptomycin mixture (P4333, Sigma-Aldrich, Castle Hill, NSW, Australia), and supplemented with 1% Newborn Calf Serum (NCS) (N4637, Sigma-Aldrich, Castle Hill, NSW, Australia). OKF6 cells were incubated at 37 °C, 5% CO_2_ for 5 to 7 days to reach 80% confluency. Epithelial cells grown to 80% confluency were subsequently detached via a pretreatment of 10 mM EDTA for 10 min, followed subsequently with incubation with a 0.25% trypsin in 1 mM EDTA solution (T4049, Sigma-Aldrich, Castle Hill, NSW, Australia) for 5 min. The viability of the keratinocytes was confirmed by trypan blue exclusion while passaging cells and during the seeding phase of each cell culture experiment (trypan blue dye, 0.4% solution, 1450021, Bio-Rad, Hercules, CA, USA).

### 2.3. Anticoagulants and 5-FU

Warfarin (A2250-10G, Sigma-Aldrich, Castle Hill, NSW, Australia), Enoxaparin sodium (E0180000, Sigma-Aldrich, Castle Hill, NSW, Australia), Edoxaban (E555520, Sapphire Bioscience, Redfern, NSW, Australia), Apixaban (15427, Sapphire Bioscience, Redfern, NSW, Australia), Dabigatran (ML2370-25MG, Sigma-Aldrich, Castle Hill, NSW, Australia), Rivaroxaban R-enantiomer (A14979, Sapphire Bioscience, Redfern, NSW, Australia), and 5-FU (F6627-5G, Sigma-Aldrich, Castle Hill, NSW, Australia), were reconstituted in acetone, filtered water, methanol, Dimethyl Sulfoxide (DMSO), HCl, DMSO, and DMSO, respectively. 

### 2.4. Proliferation Assays

All the cell lines were tested with the chemotherapeutic agent 5-FU (0.2/1/5/10 μg/mL), as well as the two conventional anticoagulants warfarin (1/5/10/20 μM) and heparin (5/20/80 U), and four new oral anticoagulants, dabigatran (5/10/20 μM), rivaroxaban (5/10/20 μM), apixaban (0.1/1/5 μg/mL), and edoxaban (5/10/20 μM). H357, H400, and OKF6 cells were seeded in 96-well plates (Corning^®^ 3596, Corning, NY, USA) under standard conditions and treated with a combination of anticoagulant and/or chemotherapeutic agents at therapeutic concentrations at T = 0 [8,11,12]. Cell viability was assessed at 0, 24, 48, and 72 h with the Cell Titer 96 Aqueous MTS assay kit (Promega Corp., Madison, WI, USA). H357 and H400 cells were seeded in 96-well plates at 5000 cells/well in 100 μL of culture medium, while OKF6 keratinocytes were seeded in 96-well plates at 8000 cells/well in 100 μL of culture medium. The choice of different initial cell seeding densities was justified by the very different mitotic rates of cancer cell lines and normal keratinocytes used for our experiments. Cells were incubated for an overnight period and then they received one wash with 100 μL of PBS before being incubated with culture medium, anticoagulants, and/or 5-FU. At each timepoint, 20 μL of MTS dye was added to each well, and the plates were incubated for 2 h at 37 °C. Absorbance readings were made at 490 nm wavelength using an automated plate reader (800 TS absorbance reader, BioTek, Currumbin, QLD, Australia). All the proliferation assay experiments were performed in triplicate.

### 2.5. Wound Healing Assays

To assess cell migration, 85–90% confluent H357/H400/OKF6 cells seeded in 96-well plates were pretreated with anticoagulant overnight and then scratched with the MuviCyte scratcher. Cells were then washed twice with PBS and treated with anticoagulants and/or 5-FU. Images were taken at 3 h intervals up to 18 h (MuviCyte Live-Cell Imaging Kit, Perkin Elmer, Glen Waverly, VIC, Australia) and analysed with ImageJ Software (ImageJ v. 1.50i, National Institutes of Health, Bethesda, MD, USA). Due to the overall number of drugs to be tested within these experiments, pairs of anticoagulants were tested simultaneously with a shared control sample. All the wound healing assay experiments were performed in quadruplicate.

### 2.6. Statistical Analysis 

Statistical analyses were performed using GraphPad Prism 8.0.1 (La Jolla, CA, USA). Differences between groups were analysed using the two-way ANOVA test.

## 3. Results

### 3.1. Treatment with Anticoagulants Reduces OSCC Cell Proliferation In Vitro

The conventional anticoagulants tested showed antiproliferative effects in the two cancer cell lines tested. Warfarin, at concentrations of 1 μM and 5 μM, significantly reduced cell growth at 72 h in H357 cell line (Figure 1). Treatment of H357 cells with 5, 20, and 80 U/mL heparin showed a significant reduction in cell growth at time 48 h (Figure 2). The same effect was seen in H400 cells at 72 h with 5 U/mL heparin (Figure 2). Both H357 and H400 cells treated with dabigatran had significantly reduced proliferation as early as 72 h (Figure 3). This effect was exerted by 5 μM and 20 μM concentrations in the H357 cell line, and for all tested concentrations for the H400 cell line. Similarly, other NOACs tested in the study reduced cancer cell proliferation. All concentrations of apixaban reduced growth of H357 and H400 cells at 72 h (Figure 4). In the H400 cell line, a significant decrease in cell vitality was observed for all tested concentrations of edoxaban at 48 h (Figure 5). At time 72, edoxaban appears to modulate the proliferation of H400, with 5 μM being inhibitory and 10 μM promoting cell proliferation. The range of concentrations of rivaroxaban tested was able to decrease OSCC cell proliferation at both 48 and 72 h, with concentrations of 10 μM in H357 (time 48 h); 5 μM in H357 and 20 μM in H400 (time 72 h) (Figure 6). Exceptions to this inhibitory trend observed for the great majority of the anticoagulants tested included warfarin, which showed a significant increase in H400 cell viability at 48 h with 1 μM and 5 μM concentrations (Figure 1); this was additionally seen at 72 h, where a dose-dependent increase of cell proliferation was observed with increasing concentration of warfarin.

To assess if the anticoagulant effects observed were specific to oral malignant cells, the normal human keratinocyte cell line OKF6 was used as normal control in this study. Similarly, all the anticoagulants except heparin exerted antiproliferative effects in OKF6 cell line. This was seen with warfarin at 5 μM (time 48 and 72 h); 5 μM and 20 μM dabigatran at time points 72 and 48 h, respectively; 10 μM rivaroxaban at time 72 h; 0.1 μg/mL apixaban at time 72 h. Interestingly, edoxaban appeared to have no effects on OKF6 cell growth, even though it has to be noted that the drug solvent for edoxaban, 0.04% methanol, showed cytotoxic effects on OKF6 as early as 24 h.

### 3.2. Anticoagulants Affect OSCC Cell Ability to Resist Chemotherapeutic Agent 5-FU

The effect of anticoagulants on chemotherapy-induced cytotoxicity was assessed overtime in OSCC cells. The majority of anticoagulants either combated the chemocytotoxicity of 5-FU or appeared to not interact with 5-FU at all. In total, 5 and 10 μM concentrations of warfarin tested reduced the chemocytotoxicity of 5-FU in H357 cells at time 24 h. All concentrations of apixaban in H357 cells reduced the cytotoxicity of 0.2 μg/mL 5-FU at 72 h. With H400 cells, 20 U, 80 U heparin, and 5 μM dabigatran were able to decrease the effectiveness of 0.2 μg/mL 5-FU at 72 h. Edoxaban also showed potential to decrease cytotoxicity of 5-FU at 72 h. Across both cell lines, 0.2 μg/mL 5-FU alone caused significant inhibition of proliferation compared to its control (*p* ≤ 0.0001), but with the addition of 5 μg/mL and 20 μg/mL in H357; and with 20 μg/mL in H400, the growth was comparable with controls again, masking the antiproliferative effect of 5-FU. On the other hand, warfarin, edoxaban, and apixaban appear to not interact with 5-FU in H400 cell lines, and heparin, dabigatran, and rivaroxaban did not interfere with 5-FU effectiveness in H357 cell lines. Comparing the cancer cell lines with OKF6 control, all concentrations of warfarin and 10 and 20 μM rivaroxaban negated the chemocytotoxicity of 0.2 μg/mL 5-FU at time 72. All concentrations of heparin, 10 and 20 μM rivaroxaban, 0.1 μg/mL apixaban, and 10 μg/mL edoxaban reduced the effectiveness of 1 μM 5-FU at 72 h. Conversely, 10 μM dabigatran was able to act synergistically with 0.2 μg/mL 5-FU at 72 h to reduce OKF6 cell proliferation.

### 3.3. Treatment with Anticoagulants Induces a Reduction in OSCC Cell Migration

Conventional anticoagulants significantly impaired cancer cell migration. Addition of all concentrations of warfarin attenuated the ability of H400 cells to migrate (Figure 7). However, this dramatic effect is not seen in H357, where only 10 μM warfarin in combination with 0.2 μg/mL 5-FU lowered the wound healing rate (time 12 h), with no inhibition of final closure at time 18 h. Heparin alone did not affect the migratory properties of H400 cells, but in combination with both 0.2 μg/mL and 5 μg/mL 5-FU, 80 U heparin reduced the rate as well as inhibited maximum closure of wound up until time 18 h (Figure 8). This inhibitory property is more pronounced in H357 cell lines, where both 40 U and 80 U heparin inhibited maximum closure of wound by around 50% at 18 h where the controls had achieved full closure of wound. Meanwhile, dabigatran did not influence the migratory properties of both H357 and H400 cancer cell lines (Figure 9). In contrast, all the other NOACs did not influence the migratory phenotype of H357 cancer cells. A total of 0.1 μg/mL apixaban and 20 μg/mL edoxaban promoted more rapid wound closure of H400 compared to control at time 6 and 9, but this effect was lost as time progressed (Figure 10 and Figure 11). With cotreatment of 5 μg/mL 5-FU, 0.1 μg/mL apixaban (Figure 10) and 5 μg/mL edoxaban (Figure 11) slowed migration of H400 cells compared to 5 μg/mL 5-FU alone at time 6, 9, and time point 9, respectively. In H400 cell line, rivaroxaban alone did not affect the migration, but 20 μM rivaroxaban combined with 5 μg/mL 5-FU slowed the migration of H400 when 5-FU alone failed to reduce the rate of H400 cell migration (Figure 12). In normal keratinocytes, conventional anticoagulants also reduced their migratory phenotype. There is no direct effect of warfarin and heparin on rate of wound closure in OKF6 cell lines (Figure 7 and Figure 8). However, 20 μM warfarin and all concentration of heparin, when combined with 0.2 μg/mL 5-FU, slowed closure rate compared to 5-FU alone. On the other hand, 20 μM dabigatran promoted migration in cells treated with 5 μg/mL 5-FU at time 6–12, with the maximum closure of wound non-significant compared to control (Figure 9). All NOACs, namely apixaban, edoxaban, and rivaroxaban did not affect migration of normal keratinocytes, except 20 μM rivaroxaban, which promoted wound closure when OKF6 cells were cotreated with 5 μg/mL 5-FU.

A comprehensive summary of the main findings from our in vitro experiments can be found in Table 1. 

## 4. Discussion

The present study has demonstrated for the first time that various anticoagulants exert antiproliferative effects on OSCC cell lines. In our experimental conditions, warfarin reduced proliferation of the H357 cancer cell line. Similarly, a study by Kirane et al. observed the antiproliferative effects of warfarin in pancreatic cancer cell lines [13]. Bai et al. have shown that coumarin, a constituent of warfarin, can inhibit proliferation and migration of HSC-2 OSCC cell lines [14]. Anti-adhesive effects and reduction of breast cancer metastasis in in vitro studies have also been demonstrated [15,16]. This finding is carried over in in vivo mice studies where warfarin inhibits metastasis of various epithelial cell cancers including breast, pancreatic, and lung cancers [15,16,17]. Observational human studies also support the finding that warfarin use has antitumourigenic properties and reduced incidence for lung, prostate, colon, and breast cancers [17,18]. The present study’s finding is concordant with the literature, where warfarin significantly inhibited migration of H357 cells and impaired wound healing in conjunction with 5-FU in H400 cells. However, the antimigratory effect is exclusive to cancer cell lines as it was absent in normal oral keratinocytes. These findings have salient clinical implications in that they suggest that the effects of warfarin are specific to oral malignant cells. The postulated mechanism of warfarin’s antiproliferative and antimigratory abilities is attributed to the inhibition of γ-carboxylation of the Gla-domain on GAS6, leading to a diminished receptor activation and reduced Axl signalling. This pathway is known in OSCC cell lines for the promotion of cancer growth and invasiveness [13,19,20,21]. Interestingly, warfarin is able to accomplish this at concentrations lower than those needed for anticoagulation [22]. Additionally, there is no established literature regarding warfarin being proproliferative toward OSCC cells. The contrasting result in the present study with the H400 cell line therefore needs further verification. The present study is also the first to demonstrate that selective concentrations of warfarin may inhibit normal keratinocyte growth, which warrants further investigation.

In vitro studies have shown that heparin and its derivatives are able to reduce tumour proliferation, migration, and invasion in epithelial cancers [23,24,25,26]. The results of the present study demonstrate that treatment with enoxaparin sodium causes reduction in proliferation of both cancer cell lines tested. The current body of literature has studied the antiproliferative effects of LMWH on melanoma cells, pancreatic adenocarcinoma cells, lung adenocarcinoma cells, glioma cells, and human breast carcinoma cells [27,28,29,30,31,32]. However, only one study exists for human OSCC cells [7,8]. The 2020 study by Camacho-Alonso et al. supports the phenomenon of reduction in OSCC proliferation with application of low molecular weight heparin [33]. It is suggested that heparin’s antiproliferative effect is from its antiangiogenic activity by impeding thrombin generation and inhibition of tissue factor (TF) expression and fibrin formation. Other studies on OSCC and lung adenocarcinoma cells observed that enoxaparin works to reduce tumour proliferation and migration via interference of PI3k/Akt and MAPK/ERK signalling pathways [8,23]. The Akt pathway is demonstrated to be active in oral squamous cell carcinomas, and its inhibition promotes induction of apoptosis [34,35].

The present study did not observe interaction between 5-FU and heparin in H357 cell lines. Interestingly, some concentrations of heparin attenuated the effects of low-concentration 5-FU on H400 proliferation; this effect has not been previously observed and warrants further investigation. Conversely, heparin and 5-FU worked synergistically to inhibit wound closure and thus migratory capacity of both cancer cell lines. While this drug combination has not been studied in human OSCC cell lines, the combination of cisplatin and heparin has been observed to result in reduced cell viability and cell migration capacity, as well as increased apoptosis of human OSCC cells [25,33].

Dabigatran, a direct thrombin inhibitor, has very limited research in the field of OSCC. Our study is the first to identify antiproliferative effect in OSCC cell lines. However, dabigatran failed to interfere with OSCC cells’ migratory phenotype. Decreasing cell viability is a finding supported by various studies testing dabigatran in numerous breast cancer cell lines [36,37]. De Feo et al. also demonstrated that orally administered dabigatran inhibits invasiveness of breast cancer cells in vitro, as well as reducing breast cancer tumour growth and metastasis in mice models [38]. Moreover, dabigatran was able to act synergistically with chemotherapeutic agents to inhibit tumour growth and progression in multiple epithelial cancer type animal models including breast and pancreatic cancers [39,40]. Interestingly, this phenomenon was not observed in the present study. Apixaban, rivaroxaban, and edoxaban all work alone in varying concentrations to reduce the proliferative action of both H400 and H357 cancer cell lines. Mouse model studies have demonstrated varying results regarding rivaroxaban and its postulated antitumourigenic effects; three studies suggested it has no effect on tumour growth [36,41,42], while a fourth showed a broad attenuation of growth [43]. High-dose apixaban has been seen to reduce cell proliferation in colon, ovarian, and prostate cancer cell lines, supporting the findings of the present study [43]. The current body of literature lacks information on in vitro studies of edoxaban on cancer cell lines. While rivaroxaban alone did not affect migration, low-dose apixaban and high-dose edoxaban led to faster wound closure at 6 and 9 h, though this effect was negligible beyond the 12 h mark. High-dose apixaban has previously been shown to reduce migration capacity in ovarian and colon cancer cells, contrary to the results of our study [43]. It is proposed that thrombin induces tumour growth and angiogenesis by activating tumour cell adhesion to platelets, endothelial cells, and subendothelial matrix protein [44,45]. Circulating factor X, through interaction with endothelial cells, can lead to an overexpression of adhesive receptors, which can promote cancer metastasis [45]. DOACs may impede this process, hence reducing tumour survival and metastasis. Therefore, research in this area should be promoted.

Literature regarding the pharmacodynamic interactions between anticoagulants, with the exception of heparin, and chemotherapeutic agents is sparse. The majority of anticoagulants studied reduced the chemocytotoxicity of 5-FU, allowing an increase in both normal and cancer cell proliferation. This novel finding warrants further urgent research. Additional in vitro research involving the use of further chemotherapeutic agents, such as cisplatin, and translation of this research line to preclinical animal models would add breadth to the present study and consolidate its findings. Our study demonstrated that there is real space for imminent and invaluable discoveries regarding the effect of anticoagulants in OSCC, with great impact on the future pharmacological management of oral cancer patients.

## Figures and Tables

**Figure 1 biology-11-00596-f001:**
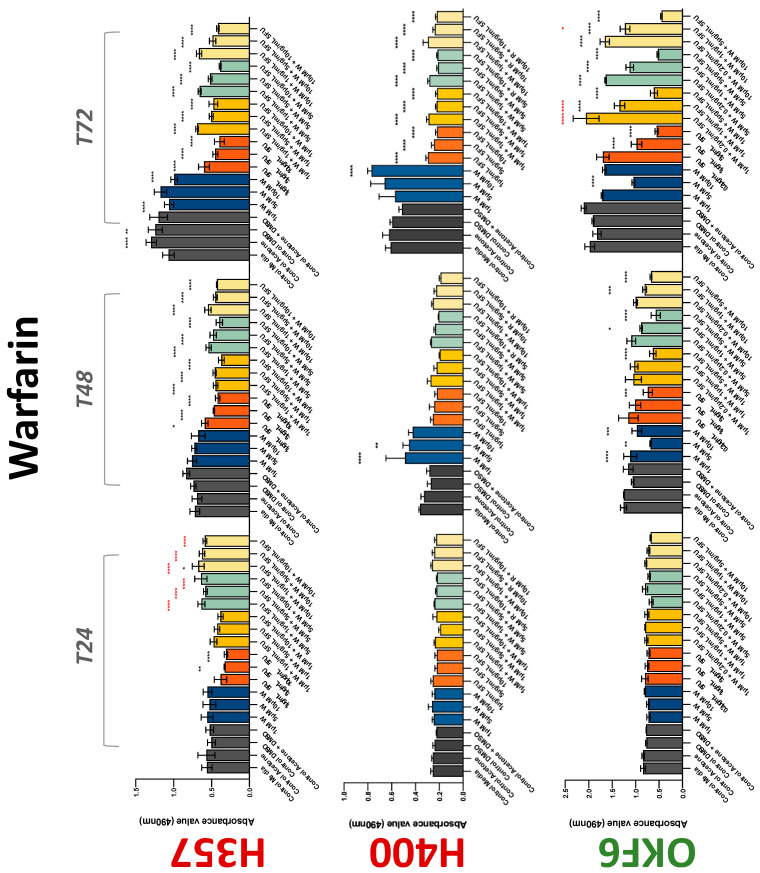
**MTS assay**. The effect of (1/5/10/20 μM) warfarin and (0.2/1/5/10 μg/mL) 5-FU on H357, H400, and OKF6 cell proliferation. Data are represented as mean ± SD. Statistical significance is given as follows: * *p* < 0.05, ** *p* < 0.01, *** *p* < 0.005, **** *p* < 0.001; * (black) compared to control group; * (red) compared to respective 5-FU groups.

**Figure 2 biology-11-00596-f002:**
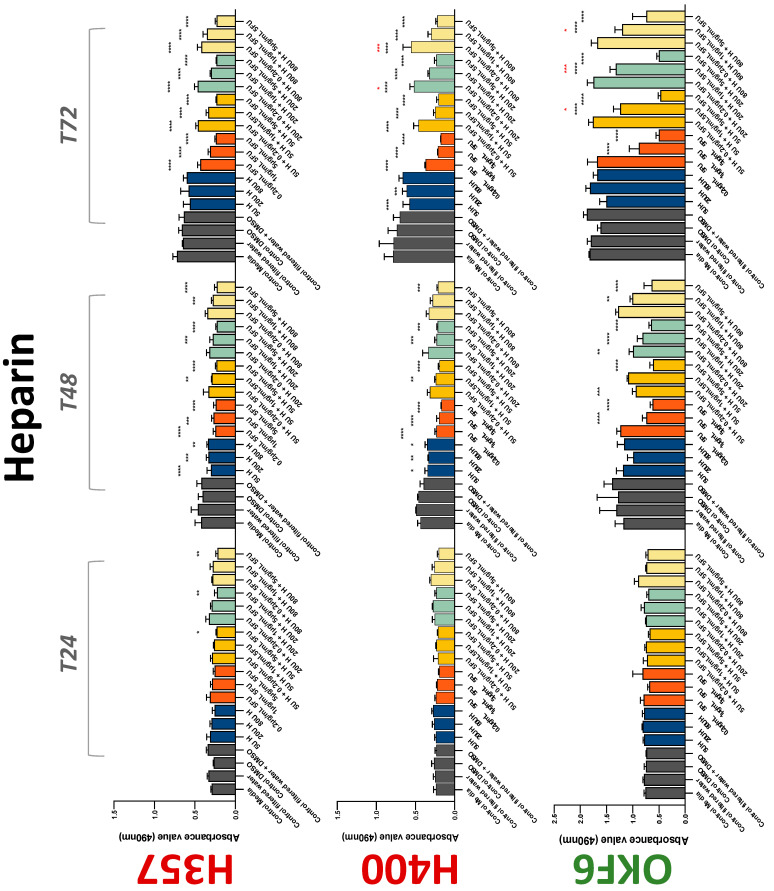
**MTS assay**. The effect of (5/20/80 U) heparin and (0.2/1/5/10 μg/mL) 5-FU on H357, H400, and OKF6 cell proliferation. Data are represented as mean ± SD. Statistical significance is given as follows: * *p* < 0.05, ** *p* < 0.01, *** *p* < 0.005, **** *p* < 0.001; * (black) compared to control group; * (red) compared to respective 5-FU groups.

**Figure 3 biology-11-00596-f003:**
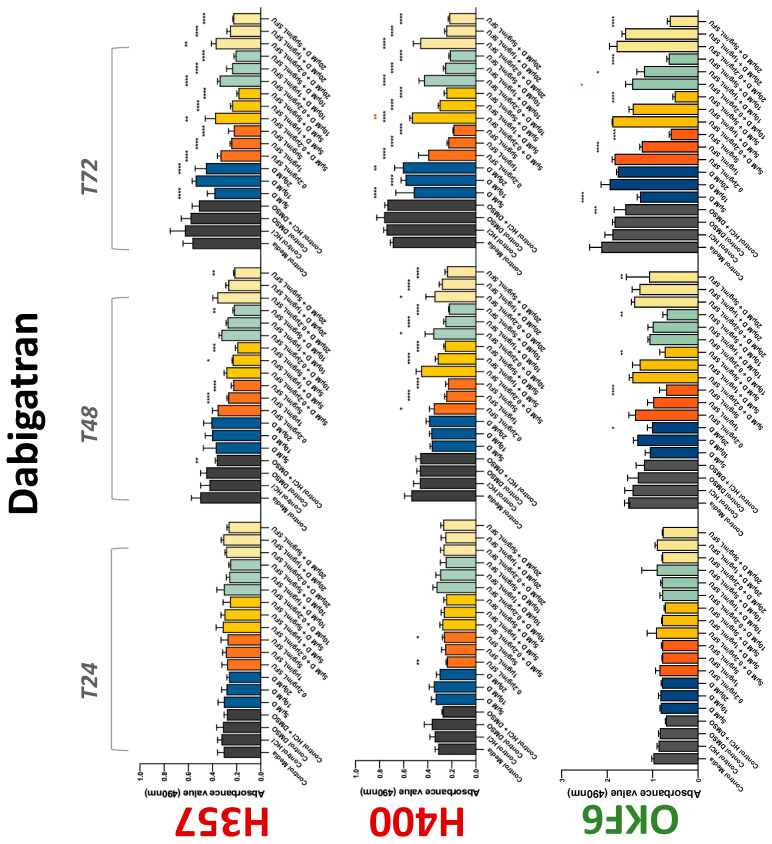
**MTS assay**. The effect of (5/10/20 μM) dabigatran and (0.2/1/5/10 μg/mL) 5-FU on H357, H400, and OKF6 cell proliferation. Data are represented as mean ± SD. Statistical significance is given as follows: * *p* < 0.05, ** *p* < 0.01, *** *p* < 0.005, **** *p* < 0.001; * (black) compared to control group; * (red) compared to respective 5-FU groups.

**Figure 4 biology-11-00596-f004:**
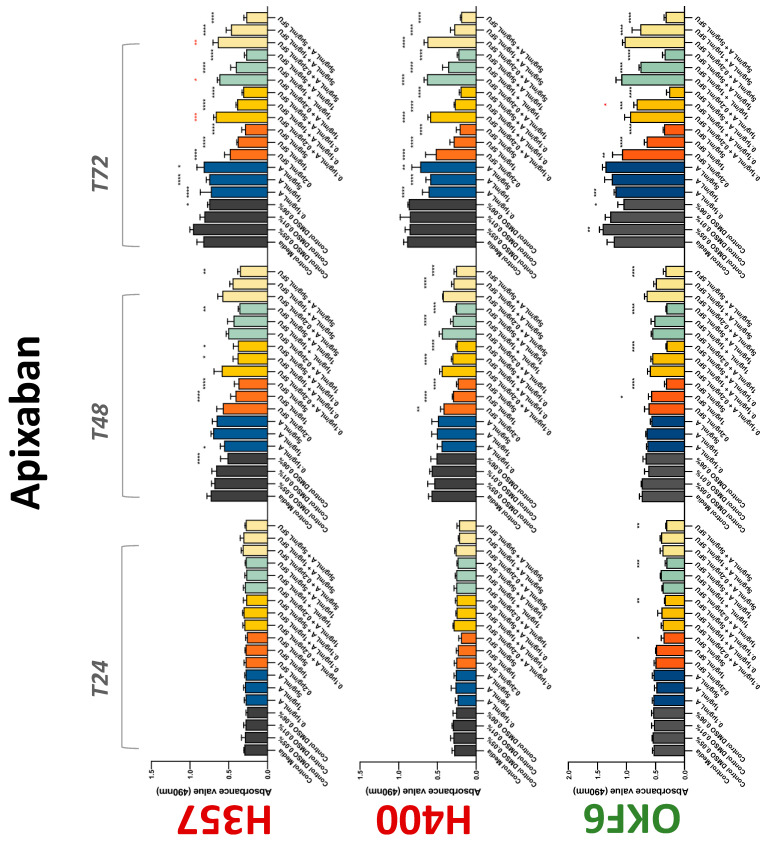
**MTS assay**. The effect of (0.1/1/5 μg/mL) apixaban and (0.2/1/5/10 μg/mL) 5-FU on H357, H400, and OKF6 cell proliferation. Data are represented as mean ± SD. Statistical significance is given as follows: * *p* < 0.05, ** *p* < 0.01, *** *p* < 0.005, **** *p* < 0.001; * (black) compared to control group; * (red) compared to respective 5-FU groups.

**Figure 5 biology-11-00596-f005:**
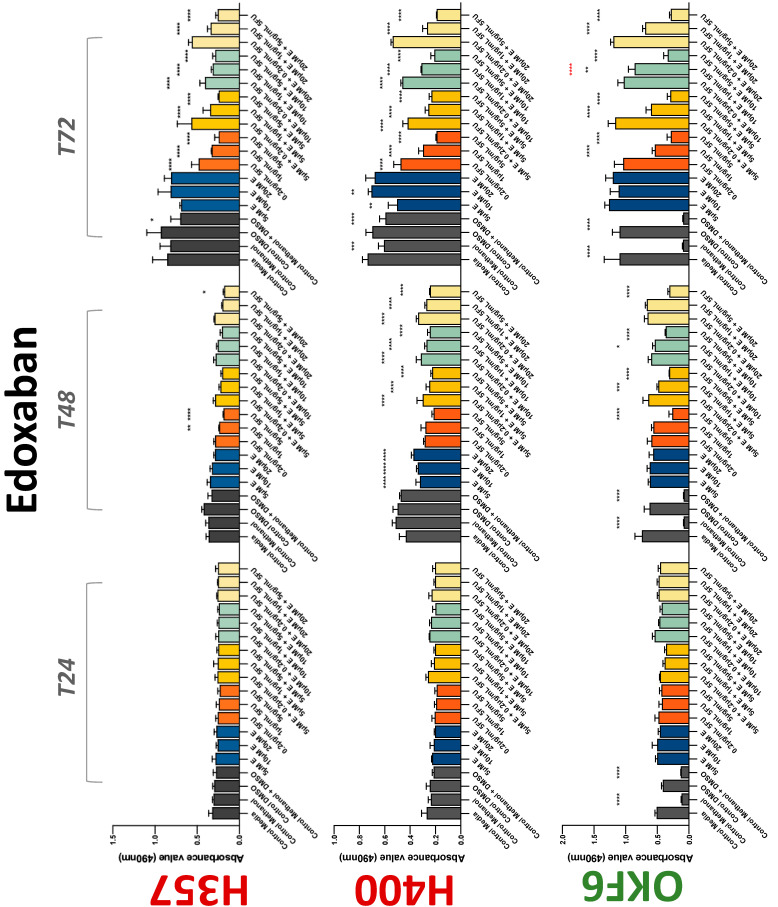
**MTS assay**. The effect of (5/10/20 μM) edoxaban and (0.2/1/5/10 μg/mL) 5-FU on H357, H400, and OKF6 cell proliferation. Data are represented as mean ± SD. Statistical significance is given as follows: * *p* < 0.05, ** *p* < 0.01, *** *p* < 0.005, **** *p* < 0.001; * (black) compared to control group; * (red) compared to respective 5-FU groups.

**Figure 6 biology-11-00596-f006:**
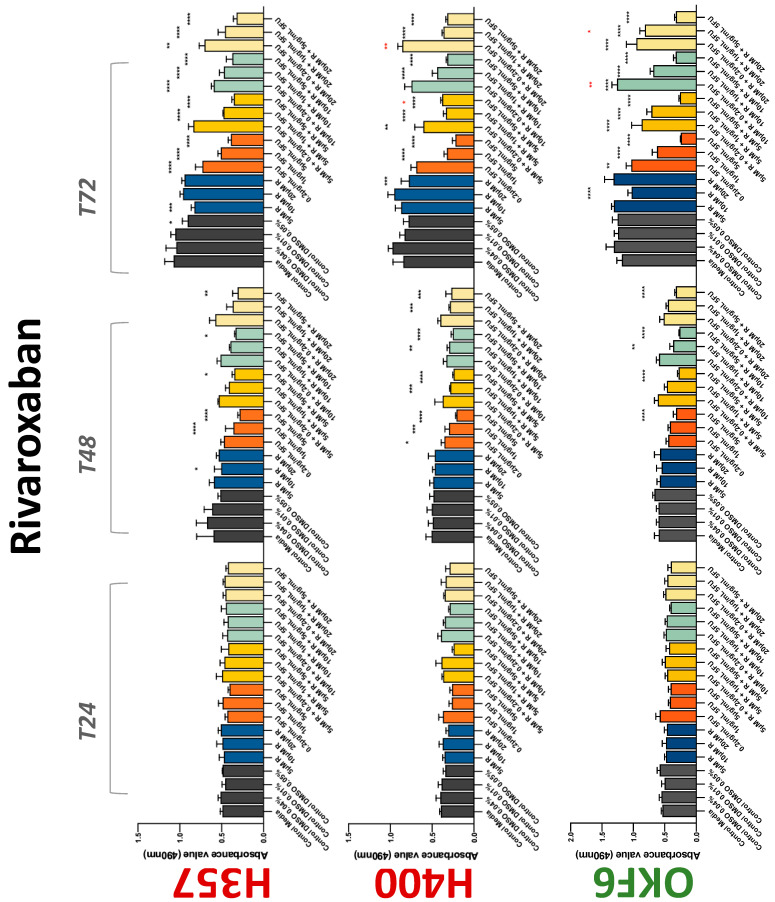
**MTS assay**. The effect of (5/10/20 μM) rivaroxaban and (0.2/1/5/10 μg/mL) 5-FU on H357, H400, and OKF6 cell proliferation. Data are represented as mean ± SD. Statistical significance is given as follows: * *p* < 0.05, ** *p* < 0.01, *** *p* < 0.005, **** *p* < 0.001; * (black) compared to control group; * (red) compared to respective 5-FU groups.

**Figure 7 biology-11-00596-f007:**
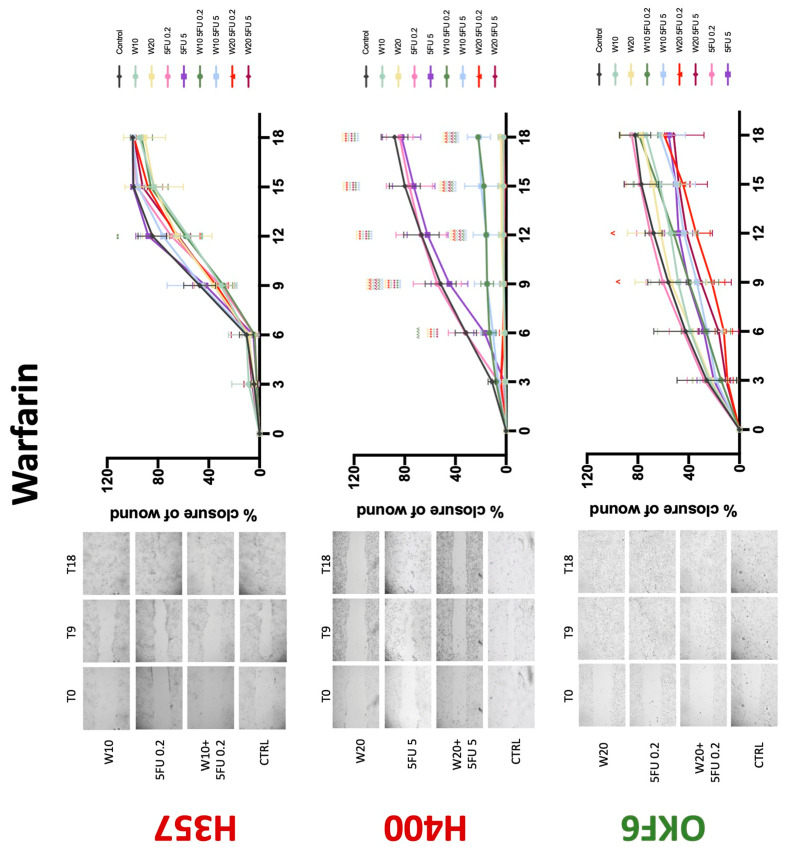
**Wound healing assay**. The effect of (10/20 μM) warfarin and (0.2/5 μg/mL) 5-FU on H357, H400, and OKF6 cell migration. Data are represented as mean ± SD. Statistical significance is given as follows: ** *p* < 0.01, **** *p* < 0.001; (*) compared to control group; ^^^^ *p* < 0.001; (^) compared to respective 5-FU groups.

**Figure 8 biology-11-00596-f008:**
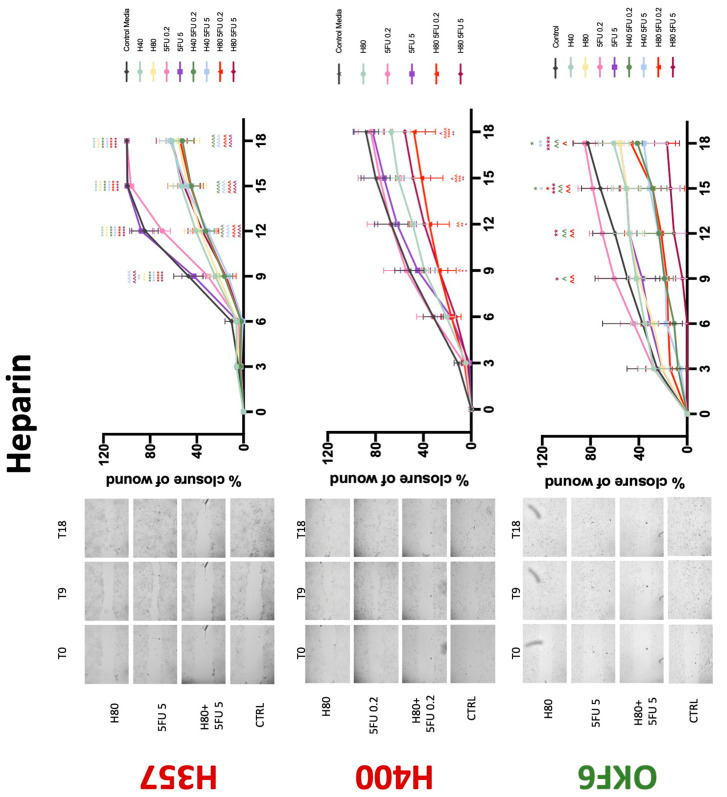
**Wound healing assay**. The effect of (40/80 U) heparin and (0.2/5 μg/mL) 5-FU on H357, H400, and OKF6 cell migration. Data are represented as mean ± SD. Statistical significance is given as follows: * *p* < 0.05, ** *p* < 0.01, *** *p* < 0.005, **** *p* < 0.001; (*) compared to control group; ^^ *p* < 0.01, ^^^ *p* < 0.005, ^^^^ *p* < 0.001; (^) compared to respective 5-FU groups.

**Figure 9 biology-11-00596-f009:**
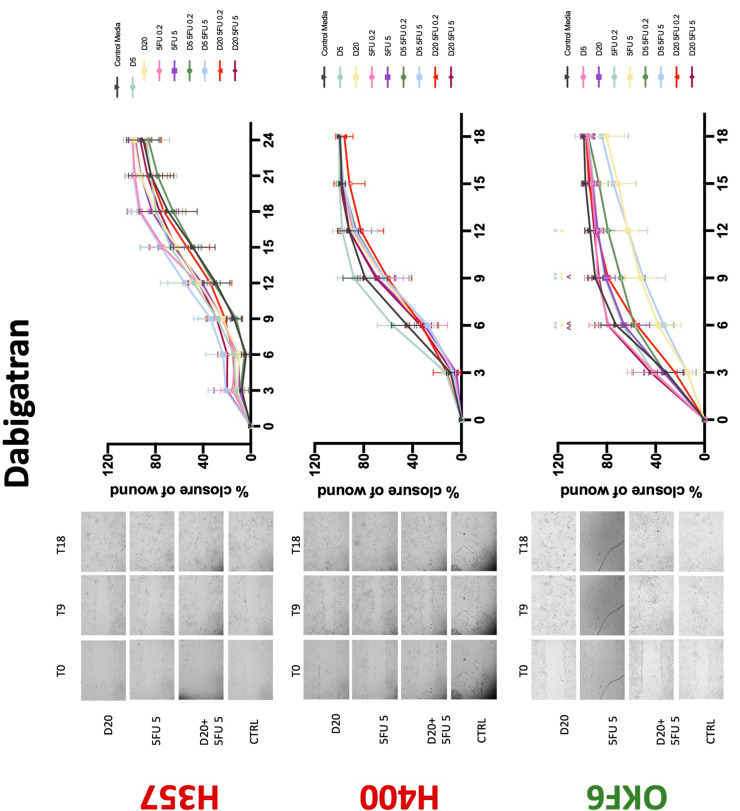
**Wound healing assay**. The effect of (5/20 μM) dabigatran and (0.2/5 μg/mL) 5-FU on H357, H400, and OKF6 cell migration. Data are represented as mean ± SD. Statistical significance is given as follows: * *p* < 0.05, ** *p* < 0.01; (*) compared to control group; ^^ *p* < 0.01; (^) compared to respective 5-FU groups.

**Figure 10 biology-11-00596-f010:**
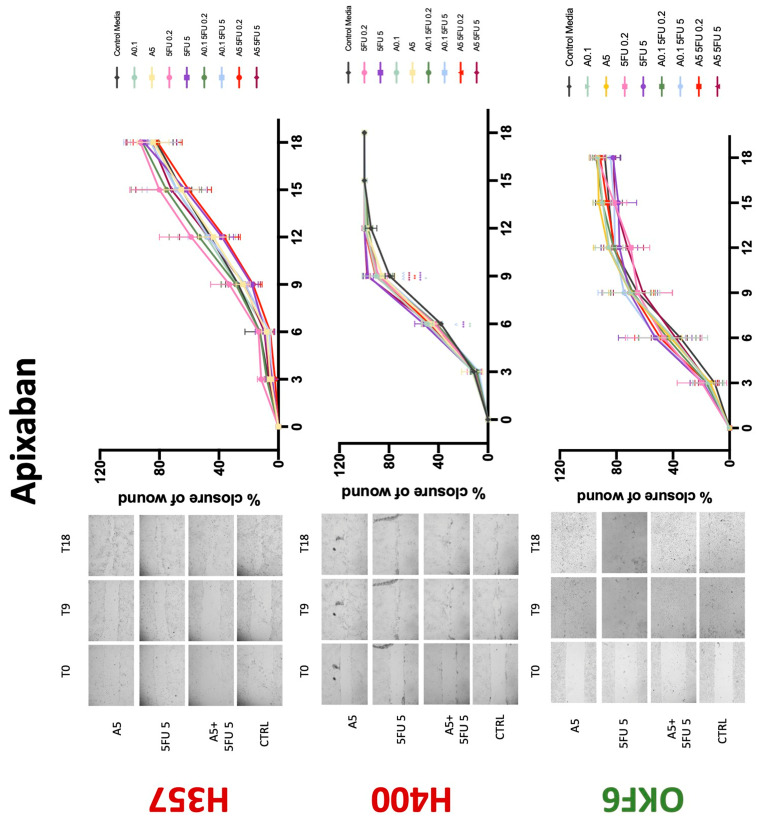
**Wound healing assay**. The effect of (0.1/5 μg/mL) apixaban and (0.2/5 μg/mL) 5-FU on H357, H400, and OKF6 cell migration. Data are represented as mean ± SD. Statistical significance is given as follows: * *p* < 0.05, ** *p* < 0.01, *** *p* < 0.005, **** *p* < 0.001; (*) compared to control group; ^^^ *p* < 0.005; (^) compared to respective 5-FU groups.

**Figure 11 biology-11-00596-f011:**
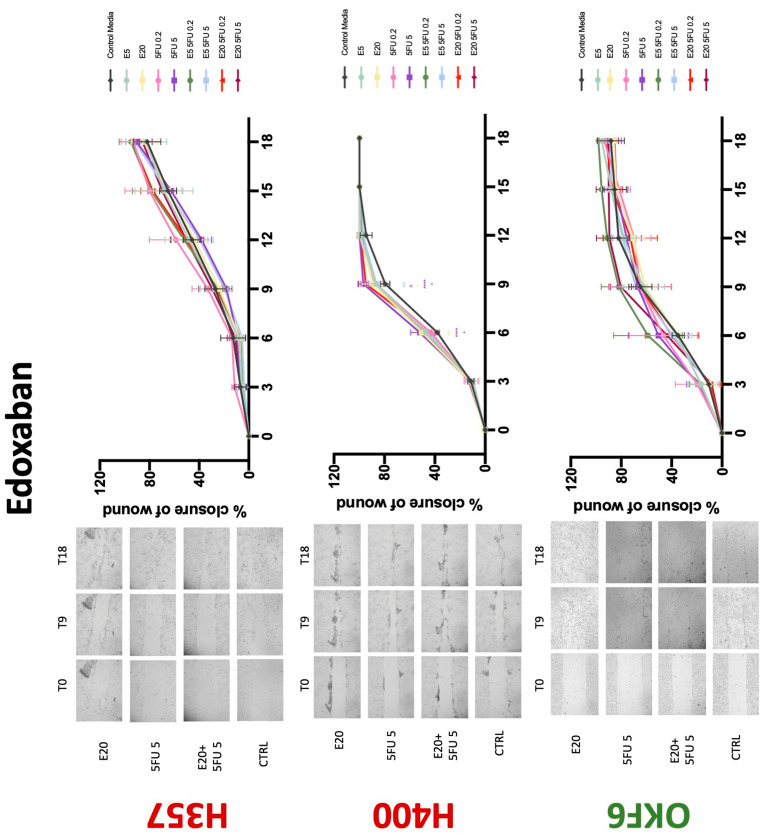
**Wound healing assay**. The effect of (5/20 μM) edoxaban and (0.2/5 μg/mL) 5-FU on H357, H400, and OKF6 cell migration. Data are represented as mean ± SD. Statistical significance is given as follows: * *p* < 0.05, ** *p* < 0.01, **** *p* < 0.001; (*) compared to control group; ^^ *p* < 0.01; (^) compared to respective 5-FU groups.

**Figure 12 biology-11-00596-f012:**
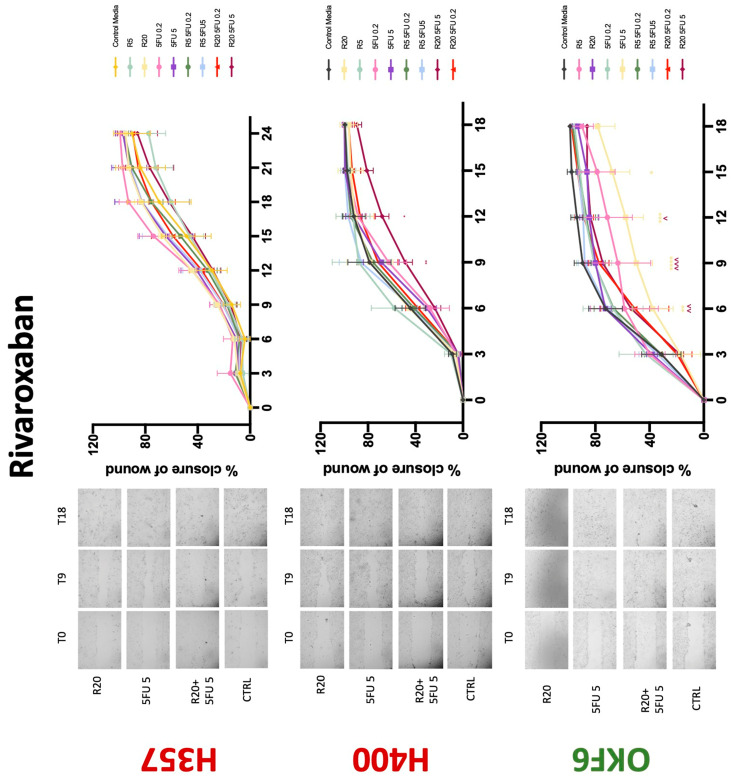
**Wound healing assay**. The effect of (5/20 μM) rivaroxaban and (0.2/5 μg/mL) 5-FU on H357, H400, and OKF6 cell migration. Data are represented as mean ± SD. Statistical significance is given as follows: * *p* < 0.05, ** *p* < 0.01, *** *p* < 0.005, **** *p* < 0.001; (*) compared to control group; ^^ *p* < 0.01, ^^^ *p* < 0.005; (^) compared to respective 5-FU groups.

**Table 1 biology-11-00596-t001:** Summary of the main in vitro findings.

	Proliferation			Migration			5-FU Effectiveness		
	H357	H400	OKF6	H357	H400	OKF6	H357	H400	OKF6
Warfarin	−(72)	+(48)	−(48)	0	−	0/−	−(24)	0	−(72)
Heparin	−(48)	−(72)	0	−	0/−	0/−	0	−(72)	−(72)
Dabigatran	−(72)	−(72)	−(48)	0	0	+	0	−(72)	+(72)
Apixaban	−(72)	−(72)	−(72)	0	+	0	−(72)	0	−(72)
Edoxaban	0	+/−(72)	0	0	+	0	−(72)	−(48)	−(72)
Rivaroxaban	−(48)	−(72)	−(72)	0	0/−	0	0	0	−(72)

Symbols: “+” = stimulates/increase; “−” = inhibits/reduces; “0” = did not affect; “()” = (hours).

## Data Availability

The data presented in this study are available on request from the corresponding author.

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
