# Peer review of "Commonly Prescribed Anticoagulants Exert Anticancer Effects in Oral Squamous Cell Carcinoma Cells In Vitro"

_biology, 2022, doi:10.3390/biology11040596_

Round 1

Reviewer 1 Report

Ling and co-workers presented a very interesting study on the effects of anticoagulants with and without 5-fluorouracil on cellular proliferation of oral squamous cell carcinomas. The simple summary and the abstract are good and the introduction is descriptive enough to read the paper. The methods and the results require a bit of editing:

Methods:

line 101: ...cell line, OKF6, were selected...   (needs a comma).

lines 128 - 129 cellular viability was confirmed by trypan blue exclusion. Is this only during cell passages? or during experiments? You also mentioned cell viability in lines 146 - 147 using the MTS assay.  Now, about this assay, it sounds like you added the MTS reagent to the wells were the cells are growing. Either clarify this or explain how the cells did not die with this.  The tetrazolium/formazan would not be part of the cell culture.

line 148 and 149: instead of "5*103", I think you mean 5 x 103 or 5,000. Please fix these numbers on both lines.  Also, please provide a justification to seed OKF6 cells at 8,000 instead of 5,000 cells/well.

Throughout the entire manuscript, microliters and milliliters should be μL and mL (capital L).

In the materials and methods, line 156, you describe "Migration assays" but in the results you refer to them as wound healing assays.  They are the same assay. Please be consistent with the assay name.

Include a "Statistical Analysis" section in the methods.

Results:

The quality of every figure is very poor. Try to enhance the quality, increase the size, reorganize the data, put it in landscape vs. portrait. The quality of the wound healing assays is even worse. It is barely legible. 

The only thing that actually saved me from being able to evaluate your paper was Table 1.  I was really happy to see that table to assess your interpretations.

I think your manuscript is great and would deserve approval for publication after addressing these points.

Good luck!

Author Response

Comments and Suggestions for Authors

Comment: Ling and co-workers presented a very interesting study on the effects of anticoagulants with and without 5-fluorouracil on cellular proliferation of oral squamous cell carcinomas. The simple summary and the abstract are good and the introduction is descriptive enough to read the paper. The methods and the results require a bit of editing.

Reply: Thank you for this comment. We are very glad this reviewer found the manuscript valuable and easy to read.

Comment: Methods: line 101: ...cell line, OKF6, were selected...   (needs a comma).

Reply: Thank you for this comment. The comma was added as suggested.

Comment: lines 128 - 129 cellular viability was confirmed by trypan blue exclusion. Is this only during cell passages? or during experiments?

Reply: Thank you for this comment. The cell viability was assessed during each cell passage, and for each experiment during the cell seeding phase. This detail has now been added to the text clearly, as per your suggestion.

Comment: You also mentioned cell viability in lines 146 - 147 using the MTS assay.  Now, about this assay, it sounds like you added the MTS reagent to the wells were the cells are growing. Either clarify this or explain how the cells did not die with this.  The tetrazolium/formazan would not be part of the cell culture.

Reply: Thank you for this comment. The MTS assay is a well-established colorimetric assay to assess cell proliferation, and it requires the addition of the dye to each experimental well, followed by a variable incubation period (1-4 hours). What makes this assay suitable for the stated purpose is the addition of the dye only once the specified timepoint has been reached. Just a few lines below (line 153) these authors already stated “At each timepoint, 20μl of MTS dye were added to each well, and the plates were incubated for 2 h at 37°C”.

Comment: line 148 and 149: instead of "5*103", I think you mean 5 x 103 or 5,000. Please fix these numbers on both lines.  Also, please provide a justification to seed OKF6 cells at 8,000 instead of 5,000 cells/well.

Reply: Thank you for this valuable comment. The symbols and the number have been corrected according to your suggestion. A justification for the cell seeding values has also been added to the text as suggested (line 151).

Comment: Throughout the entire manuscript, microliters and milliliters should be μL and mL (capital L).

Reply: Thank you for this comment. We corrected the typos throughout the manuscript.

Comment: In the materials and methods, line 156, you describe "Migration assays" but in the results you refer to them as wound healing assays.  They are the same assay. Please be consistent with the assay name.

Reply: Thank you for this comment. The assay name was made consistent across the whole manuscript, as suggested.

Comment: Include a "Statistical Analysis" section in the methods.

Reply: Thank you for this comment. A “Statistical Analysis” section has been added as suggested.

Comment: Results: The quality of every figure is very poor. Try to enhance the quality, increase the size, reorganize the data, put it in landscape vs. portrait. The quality of the wound healing assays is even worse. It is barely legible. 

Reply: Thank you for this valuable comment. We totally agree with this reviewer that the quality of the figures was very debatable. We improved all the figures of this manuscript taking into account all the provided suggestions. We increased the resolution, made sure that all the x axis data are now readable, and used landscape format. Avoiding compromises, we also made sure that no data had to be removed from the charts, e.g. comprehensive controls, etc.

Comment: The only thing that actually saved me from being able to evaluate your paper was Table 1.  I was really happy to see that table to assess your interpretations. I think your manuscript is great and would deserve approval for publication after addressing these points. Good luck!

Reply: Thank you for this comment. The authors are very grateful to this reviewer for the overall positive attitude and the very constructive comments provided.

Reviewer 2 Report

Anticoagulants are widely administred medication and cancer patients often take these drugs. In this regard the authors aimed to investigate in vitro the effects 31 of anticoagulants on OSCC cell lines and their interactions with the drug 5-FU. I found this topic really interesting for modern research. The manuscript is well written and presented and the methodology is consistent with the objectives of the study. I have only some minor suggestions to improve the quality of the manuscript:

ABTRACT

  • “MTS” abbreviation should be specified. Furthermore “5-FU” (line 32) has not been explained, whereas in line 34 it has been written as “5-fluorouracil”.

INTRODUCTION

  • The authors should better specify some epidemiological and diagnostic aspects of OSCC such as the median survival rate at 5 years and the negative prognosis for diagnostic delay. This further makes the reader understand the importance of research in perfecting the therapeutic and diagnostic strategies of this disease.

RESULTS

  • Is it possible to improve the quality of Figure 1-6? it is difficult to read some words.

Aside from these minor changes, I believe the manuscript may be accepted for publication.

Author Response

Comment: Anticoagulants are widely administred medication and cancer patients often take these drugs. In this regard the authors aimed to investigate in vitro the effects 31 of anticoagulants on OSCC cell lines and their interactions with the drug 5-FU. I found this topic really interesting for modern research. The manuscript is well written and presented and the methodology is consistent with the objectives of the study. I have only some minor suggestions to improve the quality of the manuscript:

Reply: Thank you for these very positive comments.

Comment: ABTRACT. “MTS” abbreviation should be specified. Furthermore “5-FU” (line 32) has not been explained, whereas in line 34 it has been written as “5-fluorouracil”.

Reply: Thank you for this comment. While the MTS assay is a tetrazolium-based compound and a more specific compound name of its active ingredient is: “3-(4,5-dimethylthiazol-2-yl)-5-(3-carboxymethoxyphenyl)-2-(4-sulfophenyl)-2H-tetrazolium”, the word “MTS” actually it’s still the non-abbreviated name of this technique and therefore we have to leave it as “MTS”. We corrected instead the issue related to first spelling out of 5-FU.

Comment: INTRODUCTION: The authors should better specify some epidemiological and diagnostic aspects of OSCC such as the median survival rate at 5 years and the negative prognosis for diagnostic delay. This further makes the reader understand the importance of research in perfecting the therapeutic and diagnostic strategies of this disease.

Reply: Thank you for this valuable comment. We updated the introduction section according to all these comments, adding a dedicated paragraph (line 55-63).

Comment: RESULTS: Is it possible to improve the quality of Figure 1-6? it is difficult to read some words.

Reply: Thank you for this valuable comment. We totally agree that the quality of the figures was very debatable. We improved all the figures of this manuscript also according to similar requests from the Reviewer n.1 and the AE.

Comment: Aside from these minor changes, I believe the manuscript may be accepted for publication.

Reply: Thank you for this precious comment. The authors are very grateful to this reviewer for the overall positive attitude and the very constructive comments provided.

Round 2

Reviewer 1 Report

Thank you for addressing the concerns I brought to your attention. The manuscript is in much better shape now.  I recommend it for publication.

Best of luck!

Author Response

The authors would like to sincerely thank this reviewer for the constructive feedback received and for the opportunity provided.